# Implication of lncRNA MSTRG.81401 in Hippocampal Pyroptosis Induced by P2X7 Receptor in Type 2 Diabetic Rats with Neuropathic Pain Combined with Depression

**DOI:** 10.3390/ijms25021186

**Published:** 2024-01-18

**Authors:** Ting Zhan, Shanshan Tang, Junpei Du, Jingshuang Liu, Bodong Yu, Yuxin Yang, Yuting Xie, Yanting Qiu, Guodong Li, Yun Gao

**Affiliations:** 1Department of Physiology, Basic Medical College, Nanchang University, Nanchang 330006, China; zhanting23@hotmail.com (T.Z.); tangshanshan0420@hotmail.com (S.T.); junpeidu@hotmail.com (J.D.); yangyuxin166@hotmail.com (Y.Y.); xieyuting011@hotmail.com (Y.X.); qiuyanting06@hotmail.com (Y.Q.); gc77li@hotmail.com (G.L.); 2Joint Program of Nanchang University and Queen Mary University of London, Nanchang University, Nanchang 330006, China; jp4217120085@qmul.ac.uk; 3Second Clinical Medical College, Nanchang University, Nanchang 330006, China; yubodong123@hotmail.com; 4Jiangxi Provincial Key Laboratory of Autonomic Nervous Function and Disease, Nanchang 330006, China

**Keywords:** diabetic neuropathologic pain, major depressive disorder, lncRNA, P2X7 receptor, pyroptosis, hippocampus

## Abstract

Major depressive disorder (MDD) is a common complication of diabetes and is often observed alongside diabetic neuropathic pain (DNP) as a comorbidity in diabetic patients. Long non-coding RNA (lncRNA) plays an important role in various pathophysiological processes. The P2X7 receptor is responsible for triggering inflammatory responses, such as pyroptosis, linked to pain and depression. The aim of this study was to investigate the effect of lncRNA MSTRG.81401 on hippocampal pyroptosis induced by the P2X7 receptor in diabetic rats with DNP combined with MDD (DNP + MDD). Our results showed that the expression of lncRNA MSTRG.81401 was significantly elevated in the hippocampus of DNP + MDD rats compared with the control group. Following the administration of shRNA targeting lncRNA MSTRG.81401, a notable elevation in mechanical and thermal pain thresholds was observed in rats with comorbid DNP and MDD. Additionally, significant improvements in depression-like behaviors were evident in the open-field test (OFT), sucrose preference test (SPT), and forced swim test (FST). In the DNP + MDD rats, elevated levels in hippocampal P2X7 receptor mRNA and protein were observed, along with increased co-expression of P2X7 and the astrocytic marker glial fibrillary acidic protein (GFAP). Meanwhile, in DNP + MDD rats, the heightened mRNA expression of NOD-like receptor protein 3 (NLRP3), apoptosis-associated speck-like protein (ASC), pyroptosis-related protein Gasdermin D (GSDMD), caspase-1, IL-1β, IL-18, and TNF-α was detected, in addition to increased serum levels of IL-1β, IL-18 and TNF-α. After shRNA treatment with lncRNA MSTRG.81401, the above abnormal changes in indicators for pyroptosis and inflammation were improved. Therefore, our study demonstrates that shRNA of lncRNA MSTRG.81401 can alleviate the pain and depression-like behaviors in diabetic rats associated with the comorbidity of DNP and MDD by inhibiting the hippocampal P2X7 receptor-mediated pyroptosis pathway and pro-inflammatory responses. This suggests that the P2X7R/NLRP3/caspase-1 implicated pyroptosis and inflammatory scenario may serve as a potential target for the management of comorbid DNP and MDD in diabetes.

## 1. Introduction

The global prevalence of diabetes is high and is considered to be the major epidemic of the 21st century, with an estimated global prevalence of 8.5% and increasing year by year [1]. One of the most common chronic complications of diabetes is diabetic peripheral neuropathy (DPN), which is defined as the development of symptoms associated with peripheral nerve dysfunction in patients with diabetes [2]. Approximately one-third of patients with diabetic neuropathy report intermittent or persistent sensory abnormalities (such as tingling) and pain, known as diabetic neuropathic pain (DNP) [3,4]. DNP significantly affects the quality of life of patients, but its pathogenesis remains unclear. Currently, apart from duloxetine and gabapentin with limited effectiveness, there is no specific therapy for alleviating DNP [5].

Diabetic patients often exhibit a range of psychological impediments. Research has indicated a higher propensity for depression among diabetic patients when compared to non-diabetic cohorts [6]. The prevalence of depression among diabetic patients is approximately twice that of non-diabetic individuals [7]. Around 34% of women and 23% of men with type 2 diabetes concurrently suffer from depression [8]. Furthermore, depression-associated behavioral factors not only contribute to suboptimal self-care behaviors among diabetic patients but also promote obesity and insulin resistance, thereby facilitating the development of type 2 diabetes [9,10]. Diabetic patients with neuropathic pain are particularly susceptible to comorbid depression, which in turn exacerbates their pain symptoms, establishing a pernicious cycle that significantly impacts their quality of life [11].

Pyroptosis is an inflammatory programmed cell death distinct from apoptosis. Pyroptosis may be implicated in a range of diseases, including diabetes, neurodegenerative diseases, autoimmune diseases, and cardiovascular diseases [12]. Nerve injury and neuralgia are common disorders caused by neuroinflammation. Numerous studies have shown that activation of the NLRP3 (nod-like receptor family, pyrin domain-containing 3) inflammasome is involved in a variety of inflammatory reactions, such as neuropathic pain, depression, diabetes-associated neuroinflammation, neurodegenerative diseases, cerebral ischemia-reperfusion injury, memory, and cognitive dysfunction [13]. The inflammasome was originally proposed by Tschopp et al. [14] and is present in immune cells, microglia, and astrocytes [15]. The NLRP3 inflammasome is a complex that includes the NLRP3 protein, apoptosis-associated speck-like protein containing a CARD (ASC), and pro-caspase-1 [16]. The caspase-1-dependent formation of plasma membrane pores results in the release of pro-inflammatory cytokines, such as interleukin (IL)-1β and IL-18, leading to cellular inflammation. Emerging evidence has shown pyroptosis plays a vital role in the development of diabetes [17].

Functional impairment of the hippocampal region may underpin the comorbidity of DNP and depression, and neuroinflammation in the hippocampal region is associated with the development of these events [18]. The purinergic P2X7 receptor, an ATP-gated ion channel, exhibits widespread expression within the central nervous system [19], including the hippocampus, and is implicated in the development of various central nervous diseases. ATP serves as the sole physiological agonist for P2X7R. Under pathological conditions, such as infections or neurologic disorders, there is a significant elevation in extracellular ATP concentrations, leading to the activation of the P2X7 receptor [20]. Overstimulation of the P2X7 receptor is intricately linked with inflammatory processes and can trigger the activation of inflammasomes [21,22,23]. Research indicates that stimulation of P2X7R activates the NLRP3 inflammasome, instigating the maturation and release of IL-1β [24], and subsequently promoting the pathological process of cellular pyroptosis. 

Pyroptosis can enhance inflammation, thereby facilitating the development of chronic pain [25]. Previous studies conducted in our laboratory as well as others have provided evidence suggesting that certain plant-derived medications, such as emodin, possess the ability to alleviate neuropathic pain through the inhibition of inflammatory responses [26,27]. Additionally, oxidative stress can induce pyroptosis and promote the development of depression [28]. Oxidative stress and neuroinflammation are closely related to the occurrence of depression, and antioxidants may be an effective drug for treating major depressive disorder (MDD) [29]. For example, alpha-lipoic acid, an antioxidant that can effectively treat distal symmetric painful diabetic neuropathy, may have a potential therapeutic effect on depression [30]. Therefore, direct inhibition of neuroinflammation induced by pyroptosis may be an effective strategy for treating comorbid DNP and MDD. Currently, there is a scarcity of efficacious medications for the treatment of comorbid DNP and MDD. Therefore, it is crucial to explore novel therapeutic agents that have promising inhibitory effects on pyroptosis.

The long non-coding RNA (lncRNA) plays a role in multiple important cellular functions, including chromatin rearrangement, histone modification, alternative splicing of genes, and regulation of gene expression, thereby mediating various biological processes [31]. Studies have revealed a correlation between alterations in lncRNA expression and poorly controlled blood glucose, insulin resistance, accelerated cellular aging, and inflammation in patients with diabetes [32]. LncRNA serves as a crucial epigenetic regulatory factor and plays diverse roles in many aspects of gene regulation. Recent data indicate that certain lncRNAs are dysregulated in major depressive disorder (MDD), suggesting their involvement in the pathogenesis of this disease [33]. Through high-throughput chip detection, it was observed that the expression of lncRNA MSTRG.81401 ([Gene_id: MSTRG.81401; Rat(chr17)]) is significantly upregulated in the hippocampal tissue of comorbid DNP and MDD rats. Our previous research had found that lncRNA MSTRG.81401 is associated with the pathogenesis of DNP combined with MDD [18]. The focus of the present study is to ascertain whether lncRNA MSTRG.81401 can affect cellular pyroptosis and release of inflammatory factors by influencing the expression of the P2X7 receptor in hippocampal astrocytes, which will provide a new avenue for the treatment of DNP and MDD comorbidity.

## 2. Results

### 2.1. Effect of MSTRG.81401 shRNA on Pain Behavior in Rats with Comorbid DNP and MDD

The changes in pain-related behaviors in rats were monitored using the thermal withdrawal latency (TWL) and mechanical withdrawal threshold (MWT). Compared to the control group, the TWL and MWT of rats in the DNP + MDD group were significantly decreased (*p* < 0.001). After one week of injecting shRNA against MSTRG.81401, the TWL and MWT of rats in the DNP + MDD + MSTRG.81401 shRNA group were significantly increased (*p* < 0.001) (Figure 1). The data indicate that MSTRG.81401 shRNA has the potential to alleviate neuropathic pain behaviors with associated comorbid DNP and MDD.

### 2.2. Effect of MSTRG.81401 shRNA on Depressive Behaviors in Rats with Comorbid DNP and MDD

The changes in depression-related behaviors in rats were monitored using the sucrose preference test (SPT), open-field test (OFT), and forced swim test (FST). Compared to the control group, the sucrose preference rate and total distance of locomotion in the DNP + MDD rats were significantly reduced (*p* < 0.001). Additionally, the immobility time in the FST was significantly higher in these rats (*p* < 0.001). After one week of injecting shRNA of MSTRG.81401, the SPT and OFT levels in the DNP + MDD + MSTRG.81401 shRNA group were significantly higher than those in the DNP + MDD group (*p* < 0.001). Furthermore, the immobility time in the FST was significantly reduced in the DNP + MDD + MSTRG.81401 shRNA group (*p* < 0.0001) (Figure 2). These data indicate that shRNA of MSTRG.81401 has a mitigating effect on depression-related behaviors.

### 2.3. Evaluating Expression Changes of MSTRG.81401 in Hippocampal Tissue Using Real-Time Quantitative PCR and In Situ Hybridization

The changes of MSTRG.81401 expression in the hippocampal tissue of control group and model group (DNP + MDD) rats were detected using real-time quantitative PCR and in situ hybridization techniques. The qRT-PCR results showed a significant increase in the level of MSTRG.81401 mRNA in the hippocampus of DNP + MDD rats compared to the control group (*p* < 0.001) (Figure 3A). In situ hybridization results also revealed a significant increase in the expression of MSTRG.81401 in the hippocampal tissue of DNP + MDD rats, observed in both hippocampal neurons and glial cells (Figure 3B).

### 2.4. Effects of MSTRG.81401 shRNA on Expression Levels of Hippocampal P2X7 Receptors in DNP + MDD Rat

The expression of P2X7 receptors in the rat hippocampus was assessed using real-time quantitative PCR, Western blotting, and immunofluorescence double labeling techniques. The results demonstrated that the levels of P2X7 mRNA and protein were higher in the DNP + MDD group compared to the control group. However, these changes were significantly reversed after the injection of MSTRG.81401 shRNA (*p* < 0.01) (Figure 4A–C). The co-expression of P2X7 and GFAP (a marker for astrocytes) in the hippocampus, as assessed by immunofluorescence double labeling, is significantly increased in the DNP + MDD group compared to the normal group (*p* < 0.01). However, after the injection of MSTRG.81401 shRNA, the co-expression levels of P2X7 and GFAP were significantly decreased in these rats (*p* < 0.001) (Figure 4D,E). These findings suggest that MSTRG.81401 shRNA can counteract the upregulated expression of P2X7 receptors in the hippocampus of rats with DNP and MDD comorbidity.

### 2.5. Effects of MSTRG.81401 shRNA on Pyroptosis Pathway and Inflammatory Cytokines in Hippocampal of DNP + MDD Rats

The expression levels of key molecules for inflammasome activation/proptosis (NLRP3, GSDMD, ASC, caspase-1) and pro-inflammatory cytokines (IL-1β, IL-18, and tumor necrosis factor α (TNF-α)) in the hippocampal tissue were determined using Western blotting. Compared to the control group, these proteins were significantly increased in the DNP + MDD group (*p* < 0.05). The upregulated expression levels of these proteins were significantly relieved after one week of injection with MSTRG.81401 shRNA (*p* < 0.05) (Figure 5A–M). In addition, the serum levels of the pro-inflammatory cytokines IL-1β, IL-18, and TNF-α, as measured by ELISA, were significantly increased in the DNP + MDD group compared to the control group (*p* < 0.05). Furthermore, the levels of these pro-inflammatory cytokines were significantly decreased in the DNP + MDD + MSTRG.81401 shRNA group in comparison to the DNP + MDD group (*p* < 0.01) (Figure 6A–C). These results indicate that MSTRG.81401 shRNA can inhibit pyroptosis in the hippocampus of DNP+ MDD rats and reduce the expression of pro-inflammatory cytokines.

## 3. Discussion

In the present study, we demonstrated through qPCR and in situ hybridization that the expression of hippocampal lncRNA MSTRG.81401 was significantly increased in rats with comorbid DNP and MDD, which is consistent with our previous research results [18]. Thus, lncRNA MSTRG.81401 may serve as a novel therapeutic target for treatment of diabetic comorbidity. This notion is supported by the results from administering stereotactic injection of MSTRG.81401 shRNA into the brain, since this maneuver of precise interference not only inhibited the upregulation of MSTRG.81401 expression in the hippocampus but also significantly improved both the pain behaviors and depressive behaviors in the rats with comorbid DNP and MDD. 

Studies have demonstrated that lncRNAs can participate in a wide range of biological functions, such as regulating protein expression and synthesis [34], through various molecular mechanisms, often involving interactions with one or more partners [35]. RNA-binding proteins (RBPs) are capable of binding to specific RNA molecules [36], and lncRNAs also specifically bind to RBPs and influence their associated functions [37]. The interaction relationship between proteins and RNAs is highly expansive, where each RBP can bind to multiple RNAs, and a given RNA typically interacts with multiple RBPs [38]. This relationship can be categorized into protein-focused and RNA-focused [39,40]. The former aims to identify RNAs that bind to a specific protein of interest, while the latter is to identify proteins that bind to a specific RNA [35]. Our previous work has demonstrated that the lncRNA MSTRG.81401 can regulate the expression of P2X4, potentially impacting the occurrence and development of some diseases [17]. The primary focus of the present study is to explore the potential regulatory role of lncRNA MSTRG.81401 on P2X7. The P2X7 receptor, known for its potent pro-inflammatory effects, can induce inflammasome activation [41,42,43]. It is associated with the development of various inflammation-related diseases, including diabetic complications, neurological disorders, peripheral inflammation, and cancer [44]. In this study, qRT-PCR and Western blotting have shown that shRNA of MSTRG.81401 can reduce the upregulated expression of P2X7 in the hippocampus of rats with comorbid DNP and MDD, pointing to the role of lncRNA MSTRG.81401 in the pathogenesis of these diabetic complications. 

In addition, the results of double immunofluorescence staining have revealed that the co-expression level of P2X7 and the astrocyte marker GFAP in the hippocampus of the rats with comorbid DNP and MDD is significantly increased, indicating the activation of hippocampal astrocytes in these rats. Astrocytes, ubiquitously distributed throughout the central nervous system, serve a plethora of functions, e.g., encompassing ionic and neurotransmitter homeostasis, synaptogenesis or removal, synaptic modulation, neurovascular coupling, and blood–brain barrier maintenance. Activation of astrocytes may yield deleterious consequences on neuronal functionality, implicating neurological and psychiatric disorders [45,46,47]. It is highly likely that hippocampal astrocyte activation is involved in the development of DNP and MDD in view of the essential role of the hippocampus in the control of emotion related events. Or work suggests that MSTRG.81401 may participate in the pathogenesis of comorbid DNP and MDD by upregulating the expression of P2X7 and activation of hippocampal astrocytes, since shRNA of MSTRG.81401 can reverse these changes, as demonstrated by the significant relief of elevated co-expression of P2X7 and GFAP in rats with comorbid DNP and MDD. 

The P2X7-mediated NLRP3 pathway plays a pivotal role in cognitive impairments associated with various neurodegenerative conditions, such as Alzheimer’s disease, vascular dementia, and cognitive disorders linked to diabetes [48,49,50]. Activation of P2X7 facilitates the assembly of NLRP3 inflammasomes, the intracellular multi-protein complexes that trigger inflammatory responses and pyroptotic cell death via activating the effector molecule caspase-1. Active caspase-1 subsequently augments the production of pro-inflammatory cytokines such as IL-1β and IL-18 and activates GSDMD [51]. Research has indicated that neuroinflammation can enhance the immunoreactivity of NLRP3 inflammasome/caspase-1 in astrocytes, leading to increased release of pro-inflammatory cytokines, which contribute to cell pyroptosis associated brain injury [52]. To further explore the molecular mechanisms related to the involvement of lncRNA MSTRG.81401 in the pathogenesis of DNP combined with MDD, we examined the expression levels of molecules related to pyroptosis and inflammation in rat hippocampal tissues. The results demonstrate that in rats with comorbid DNP and MDD, there are significant increases in the levels of NLRP3, ASC, and caspase-1 proteins, accompanied by a concurrent elevation in the concentrations of IL-1β, TNF-α, and IL-18 proteins within the hippocampus and serum. Moreover, the administration of MSTRG.81401 shRNA can effectively inhibit these alterations of key players implicated in the induction of pyroptosis and inflammation. Thus, these findings indicate that MSTRG.81401 shRNA may alleviate the pain and depression-like behaviors in rats with comorbid DNP and MDD by suppressing the pyroptosis-inflammation pathway in hippocampal cells through downregulating P2X7. 

In conclusion, our study revealed an elevation of lncRNA MSTRG.81401 in the hippocampus of DNP + MDD rats, which subsequently increased the expression and activation of the P2X7 receptor. This led to the activation of the NLRP3/caspase-1-mediated pyroptosis pathway and subsequent inflammation escalation. Moreover, our experiments demonstrated the use of shRNA against lncRNA MSTRG.81401 was able to suppress these processes effectively. Therefore, we propose that targeting lncRNA MSTRG.81401 could serve as a promising therapeutic approach for managing comorbid DNP and MDD. The knowledge from this work using MSTRG.81401 shRNA may provide a novel avenue to deal with the comorbidity.

## 4. Materials and Methods

### 4.1. Establishment of Animal Models

Male Sprague–Dawley rats, weighing 200 ± 10 g, were purchased from Changsha Tianqin Biotechnology Co., Changsha, China. Considering that the presence of estrogen and the estrus cycle in female mice may influence the outcomes of animal behavior tests, we specifically chose male Sprague–Dawley rats for our study. The animal experiments were approved by the Experimental Animal Welfare Ethics Committee of Nanchang University (Approval code: NCULAE-20230128089). The procured male rats were arbitrarily segregated into four groups equally, namely the standard control group (control group), DNP combined with MDD group (model group), model rats treated with MSTRG.81401 shRNA group (model + MSTRG.81401 shRNA group), and model rats treated with NC shRNA group (model + NC shRNA group). The timeline for animal model establishment is shown in Figure 7. After one week of adaptation to a normal diet, the rats were fed a high-sugar and high-fat diet for four weeks. The rat models of type 2 diabetes (fasting blood glucose > 7.8 mmol/L, postprandial blood glucose > 11.1 mmol/L, random blood glucose > 16.7 mmol/L) were induced by intraperitoneal injection of streptozotocin. DNP rats were selected based on MWT and TWL measurements, followed by four weeks of chronic unpredictable mild stress [53]. SPT, FST, and OFT were utilized to identify rats with MDD, ultimately establishing a rat model with comorbid DNP and MDD. Based on the group assignments, equivalent amounts of MSTRG.81401 shRNA or NC shRNA were injected into the lateral ventricles of the rats (with the Bregma point as the origin, X = +1.5 mm, Y = −0.8 mm, Z = −3.8 mm) using a stereotaxic apparatus. The ratio of shRNA to transfection reagent was 1:2 (4 μg:8 μL) according to the manufacturer’s instructions.

### 4.2. Assessment of Neuropathic Pain Behaviors

Mechanical withdrawal threshold (MWT). The MWT of the rats was ascertained using a BME-404 type electronic algometer (Institute of Biomedical Engineering, Chinese Academy of Medical Sciences, Tianjin, China), with a Von-Frey filament diameter of 0.6 mm. The pressure measurement range of the stimulator was 0.1 to 50 g, with a resolution of 0.05 g. After a 30-min adaptation period in a colorless and transparent glass frame for the tested rat, the central part of the left hind foot of the rat was stimulated with a Von-Frey wire. The pressure was gradually increased until the rat lifted its left hind paw, and the measurement value was recorded. The stimulation interval was 2 min, and the average of three consecutive measurements was obtained [11].

Thermal withdrawal latency (TWL). The BME-410C thermal stimulation detector mentioned above was also utilized to assess TWL. Prior to evaluation, each rat was acclimated for 30 min on a glass plate within a transparent square bottomless acrylic box. At the commencement of the experiment, a beam of radiant heat was used to stimulate the mid-surface of the hind paw plantar surface and activate a timer to start the test. The cut-off time for thermal stimulation was 30 s, and the timer was terminated at the same time when the rat lifted the hind paw and immediately disconnected the light source. The time displayed on the timer was recorded as the TWL, with the unit being s. Testing commenced every 2 min, with the final average derived from three stable data points [11].

### 4.3. Assessment of Depressive Behaviors 

Sucrose preference test (SPT). SPT is a reward-based test used as an indicator of anhedonia and depression-like behavior [18]. The test involved a 48-h sucrose drinking training for the experimental rats. For the first 24 h, each cage of rats was given two bottles of 2% sucrose solution, followed by one bottle of 2% sucrose solution and one bottle of pure water for the next 24 h (with a mid-way swap of the two bottles’ positions). Subsequently, the rats were deprived of food and water for 14–23 h, and the consumption of sucrose solution and pure water within a one-hour period was measured. Sucrose preference rate (%) = sucrose consumption/total consumption × 100% [11].

Open-field test (OFT). The OFT should be conducted in a quiet environment, minimizing human interference. The rats were placed in a dark environment for 30 min to adapt. Subsequently, the test rats were placed in a black box measuring 40 × 60 × 50 cm. Each rat was gently placed in the middle of the box, and the total distance traveled by the rat within 5 min was observed and recorded using SMART 3.0 software, measured in centimeters. Before introducing the next rat into the box, the equipment should be cleaned with a 75% ethanol solution [11].

Forced swimming test (FST). The rats are placed in a glass cylinder with a height of 80 cm and an inner diameter of 40 cm. The water temperature was approximately 20 °C, with a water depth of 30 cm. The duration of immobility of the rats in the water within a 5-min period (i.e., the time when the rats ceased struggling and floated in a fixed posture) was recorded and measured in s [11].

### 4.4. qRT-PCR

The rats were anesthetized by intraperitoneal injection of 10% chloral hydrate solution (CAS:302-17-0; Shanghai Macklin Biochemical Co., Ltd., Shanghai, China). The freshly obtained hippocampal tissue was washed with phosphate-buffered saline (PBS) buffer solution and stored in an RNA holding storage solution. After overnight incubation at 4 °C, the tissue samples were stored at −20 °C for long-term use. Prior to use, all instruments were treated with DEPC. Total RNA was isolated from the tissue samples using TRIzol Total RNA Reagent (Beijing TransGen Biotech Co., Beijing, China). After total RNA extraction and reverse transcription, PCR systems were prepared, and the amplification was detected using a StepOnePlus PCR system (Applied Biosystems, Foster City, CA, USA). The primer sequences are as follows: β-actin, forward 5′-TAA AGA CCT CTA TGC CAA CAC AGT-3′ and reverse 5′-GGG GTG TTG AAG GTC TCA AA-3′; P2X7, forward 5′-GAT GGA TGG ACC CAC AAA GT-3′, and reverse 5′-GCT TCT TTC CCT TCC TCA GC-3′; and lncRNA MSTRG.81401, forward 5′-AAGCGGCATAATCCAATGTC-3′ and reverse 5′-CATAAGCAGTTTGGGGCAAT-3′.

### 4.5. In Situ Hybridization

The in situ hybridization experiment was conducted using the RNASweAMI™ in situ hybridization DAB detection kit (Servicebio, Wuhan, China). Following the instructions, tissue paraffin sections were subjected to dewaxing and rehydration. Subsequently, a heat repair was performed using 1× repair solution, followed by the addition of Proteinase K working solution to digest the tissue for 20–30 min. After incubating at room temperature with 3% H_2_O_2_ solution for 15 min in the dark, the preheated hybridization solution was added, and the samples were incubated at 40 °C in a preheated hybridization instrument for 30 min. The hybridization solution was discarded, and the preheated target probe mixture 1 in the hybridization solution was added to cover the sample, followed by incubation in the hybridization instrument at 40 °C for 3 h. The target probe mixture 1 in hybridization solution was discarded, and the sections were sequentially washed with preheated 2 × SSC, 1 × SSC, 0.5 × SSC, and 0.1 × SSC solutions for 5 min each. The sections were gently shaken to remove excess liquid, and the preheated target probe mixture 2 in hybridization solution was added to cover the sample, followed by incubation at 40 °C for 45 min. The target probe mixture 2 in the hybridization solution was discarded, and the sections were washed using the same method as before. The sections were gently shaken to remove excess liquid, and the preheated DIG signal probe in hybridization solution was added to cover the sample, followed by incubation in the hybridization instrument at 37 °C for 45 min. The DIG signal probe in the hybridization solution was discarded, and the sections were washed using the same method as before. For HRP conjugation with digoxigenin antibody, the sections were gently shaken to remove excess liquid, and the tissue was covered with blocking solution and incubated at room temperature for 30 min. The sections were gently shaken to remove excess liquid, and the sample was covered with anti-DIG (HRP) working solution, followed by incubation at 37 °C for 50 min. After that, the sections were washed with 1 × PBS four times, each time for 5 min. For DAB staining, DAB chromogen solution was applied onto the tissue section for 3–5 min, followed by rinsing with distilled water. The section was stained with hematoxylin solution for 3 min, rinsed with tap water, and differentiated with hematoxylin differentiation solution for 3–5 s. Then the section was rinsed with tap water and treated with hematoxylin bluing solution for 3–5 s, followed by running water rinse and dehydration. For image capture, the section was observed under a standard bright-field microscope, using an appropriate magnification (20×–40×) objective lens. The target probe detection results are manifested as brown punctate or clustered signals within the cell cytoplasm, while the cell nuclei appear as deep blue.

### 4.6. Western Blotting

After anesthetizing the rats, the hippocampal tissue was isolated and rinsed with ice-cold PBS. The tissue was then placed in the spherical part of a 2 mL homogenizer and mixed with RIPA lysis buffer (C1053+; Applygen, Beijing, China), which had been prepared in advance with proportional amounts of protease inhibitor and phosphatase inhibitor. The tissue was ground on ice for approximately 30 min until a homogeneous mixture was obtained without visible clumps. It was then centrifuged at 4 °C and 15,000 rpm for 10 min, and the supernatant was transferred to a new centrifuge tube while measuring its volume. An appropriate amount of protein loading buffer (P51114; TransGen Biotech, Beijing, China) was added, and the mixed sample was boiled for 5 min. Protein separation was performed using SDS-PAGE, followed by transfer onto an immunoblotting PVDF membrane (Millipore, Burlington, MA, USA). The membrane was then blocked with 5% skim milk (1172GR500; BioFroxx, Guangzhou, China) for two hours, followed by washes with TBST. The primary antibody was then added and incubated overnight at 4 °C. Afterward, the membrane was washed three times with 1 × TBST for 10 min each. The secondary antibody (Zhong Shan-Gold Bridge, Beijing, China) was diluted 1:2000 and incubated with the membrane at room temperature for 2 h. The PVDF membrane was then washed three times for 10 min each and treated with an application of enhanced chemiluminescent substrate (FD8020; Fude Biotechnology Co., Ltd., Hangzhou, China). Visualization of bands was performed using a ChemiDoc^TM^ XRS+ system (Bio-Rad, Hercules, CA, USA). Densitometry analysis of the target bands was conducted using ImageJ 1.53e software (National Institutes of Health, Bethesda, MD, USA). The relative expression levels of target proteins were calculated using the corresponding β-actin bands as controls. The primary antibodies used and their sources are as follows: P2X7 (1:800, APR-008, Alomone, Jerusalem, Israel), NLRP3 (1:800, IMG-6668A, Novus Biologicals, Centennial, CO, USA), GSDMD (1:800, NBP2-33422, Novus Biologicals, CO, USA), ASC (1:500, DF6304, Affinity Biosciences, Cincinnati, OH, USA), pro-caspase-1 (1:1000, AB179515, Abcam, Cambridge, UK), cleaved-caspase-1 (1:1000, AF4022, Affinity Biosciences, OH, USA), IL-1β (1:500, BA14789, Boster, Pleasanton, CA, USA), TNF-α (1:500, BA0131, Boster, CA, USA), IL-18 (1:500, DF6252, Affinity Biosciences, OH, USA), and β-actin (1:1000, Zhong Shan-Gold Bridge, Beijing, China).

### 4.7. Double Immunofluorescence Labelling

Paraffin sections of hippocampus were taken for dewaxing treatment, washed with PBS, and fixed with 4% paraformaldehyde. Permeabilization was achieved by incubating the slices with 0.5% Triton-X 100 (BL-372, Sbjbio, Nanjing, China) for 15 min at 37 °C. Blocking was performed using 1-h incubation at 4 °C with goat serum (ZU-9022, Zhong Shan-Gold Bridge, Beijing, China). The primary antibodies: anti-glial fibrillary acidic protein (GFAP, 1:200, Biolegend, San Diego, CA, USA) and anti-P2X7 (1:500, Alomone Labs, Jerusalem, Israel) were incubated overnight at 4 °C. Subsequently, the secondary fluorescent antibodies: goat anti-rabbit tetramethylrhodamine (TRITC) (1:200, Zhong Shan-Gold Bridge) and goat anti-mouse fluorescein isothiocyanate (FITC) (1:200, Zhong Shan-Gold Bridge) were incubated in the dark for one hour at 37 °C. The cell nuclei were stained with DAPI (Boster, CA USA), and the slides were sealed with anti-fluorescence quencher (Boster, CA, USA). Finally, images were captured using a confocal microscope (Olympus, Tokyo, Japan). The average light density values were calculated using ImageJ software to describe the staining intensity [11].

### 4.8. Enzyme-Linked Immunosorbent Assay (ELISA)

The concentrations of IL-1β, TNF-α, and IL-18 in the serum were determined using respective ELISA kits (Wuhan Shenke Experimental Technology, Wuhan, China). According to the manufacturer’s instructions, the standard samples (1st to 5th) were first prepared by proper dilutions. A 96-well plate was set up for blanks (without samples and enzyme reagents), standards, and samples to be tested, respectively, all in triplicate. The standard samples were added in a volume of 50 μL, while the test samples were first added in 40 μL dilution solution and then supplemented with 10 μL of serum to be tested. The plate was then sealed with a sealing film and incubated at 37 °C for 30 min. Afterward, the plate was washed five times, and 50 μL of enzyme reagent was added to each well. The plate was sealed again and incubated at 37 °C for 30 min. Following another five washes, 50 μL of color reagent A and 50 μL of color reagent B were added to each well. The plate was gently shaken to ensure thorough mixing and then incubated at 37 °C in the dark for 10 min for color development. Finally, 50 μL of stop solution was added to terminate the reaction, and the absorbance (OD value) of each well was measured at 450 nm wavelength.

### 4.9. Statistical Analysis

The experimental data were analyzed using SPSS 20.0 software (SPSS, Chicago, IL, USA). The differences in behavioral data were assessed using a two-way analysis of variance (ANOVA) combined with Tukey’s post hoc test. For the remaining experimental data, a one-way ANOVA was employed, and pairwise comparisons between groups were conducted using the LSD test. All data were presented as mean ± standard error for statistical description, and a *p* < 0.05 was considered statistically significant.

## 5. Conclusions

The administration of shRNA targeting lncRNA MSTRG.81401 can inhibit the expression of hippocampal P2X7, reduce pyroptosis and inhibit the release of inflammatory cytokines, thus improving pain and depression-like behaviors in rats with comorbid DNP and MDD. The underlying molecular mechanism may involve the suppression of the activation of the P2X7/NLRP3/caspase-1 pyroptosis pathway in hippocampal cells, thereby alleviating inflammatory responses and pathological damage.

## Figures and Tables

**Figure 1 ijms-25-01186-f001:**
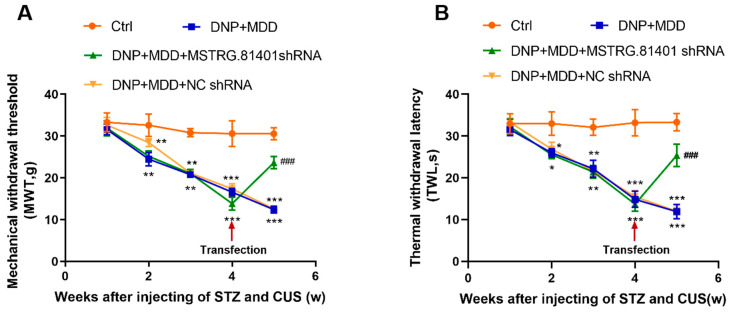
Effects of MSTRG.81401 shRNA on mechanical withdrawal threshold (MWT) (**A**), thermal withdrawal latency (TWL) (**B**) of comorbid diabetic neuropathic pain (DNP) and major depressive disorder (MDD) rats. Values are mean ± SEM, *n* =  6; * *p* <  0.05, ** *p* <  0.01 and *** *p* <  0.001 vs. Ctrl group; ### *p* < 0.001 vs. DNP + MDD group.

**Figure 2 ijms-25-01186-f002:**
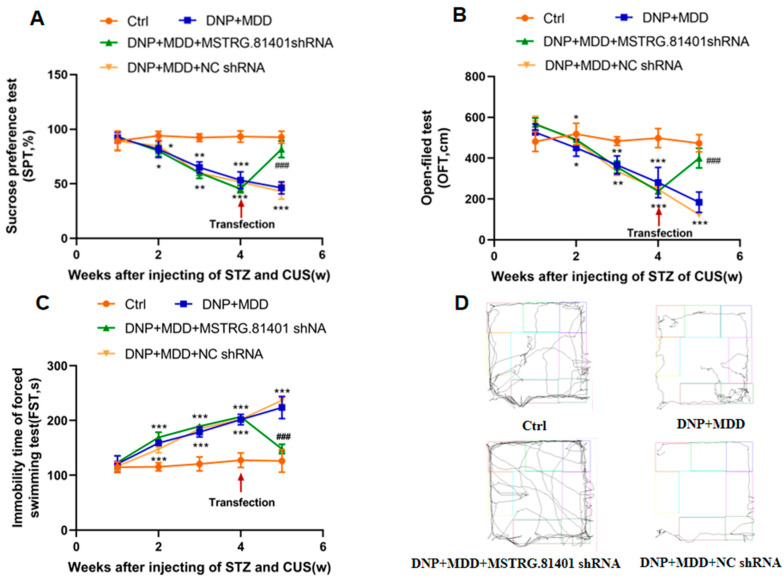
Effects of MSTRG.81401 shRNA on sucrose preference test (SPT) (**A**), open-field test (OFT) (**B**) and forced swimming test (FST) (**C**) of comorbid DNP and MDD rats; Trajectory maps show the open-field movement of rats (**D**). Values are mean ± SEM, *n*  =  6; * *p* <  0.05, ** *p* <  0.01, *** *p* <  0.001 vs. Ctrl group; ### *p* <  0.001 vs. DNP + MDD group.

**Figure 3 ijms-25-01186-f003:**
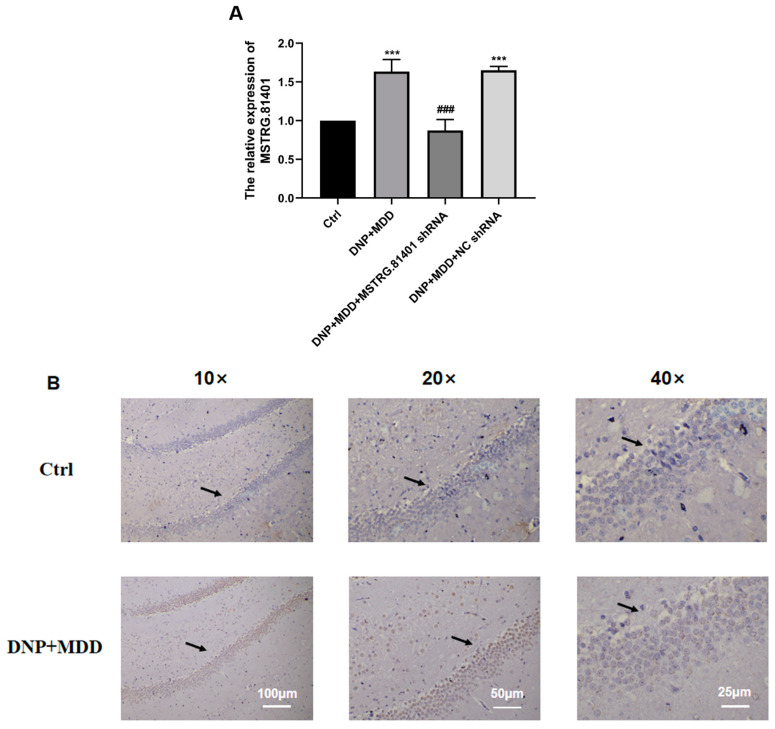
Measurement of MSTRG.81401 expression levels in hippocampus by qPCR (**A**) and in situ hybridization (**B**). Values are mean ± SEM, *n* = 6; *** *p* < 0.001 vs. Ctrl group; ### *p* < 0.001 vs. DNP + MDD group. Black arrows indicate the immunostaining cells.

**Figure 4 ijms-25-01186-f004:**
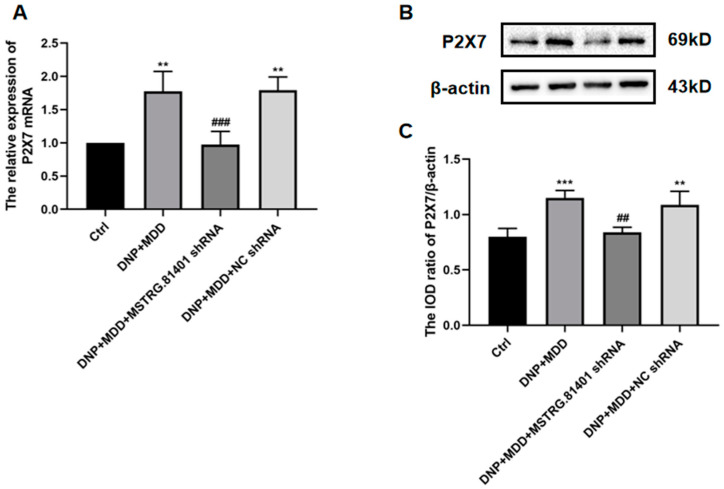
Measurements of P2X7 mRNA and protein levels in hippocampus and co-expression of P2X7 with GFAP. The levels of P2X7 mRNA were measured by qRT-PCR (**A**). P2X7 protein in hippocampus of experimental rats was assessed by Western blotting (**B**) and the analyzed results are shown in (**C**). Co-expression of GFAP and P2X7 in hippocampus was detected by immunofluorescence double labeling (**D**); Blue staining (DAPI) marks the nucleus, green staining (GFAP) marks the astrocyte, red staining marks P2X7, and the yellow signal is a combination of green and red signals. The scale bar represents 50 μm. The relative fluorescence intensity levels of GFAP and P2X7R co-expression are presented in (**E**). Values are mean ± SEM, *n* = 6; * *p* < 0.05, ** *p* < 0.01, *** *p* < 0.001 vs. Ctrl group; ## *p* < 0.01, ### *p* < 0.001 vs. DNP + MDD group. White arrows indicate the immunostaining cells.

**Figure 5 ijms-25-01186-f005:**
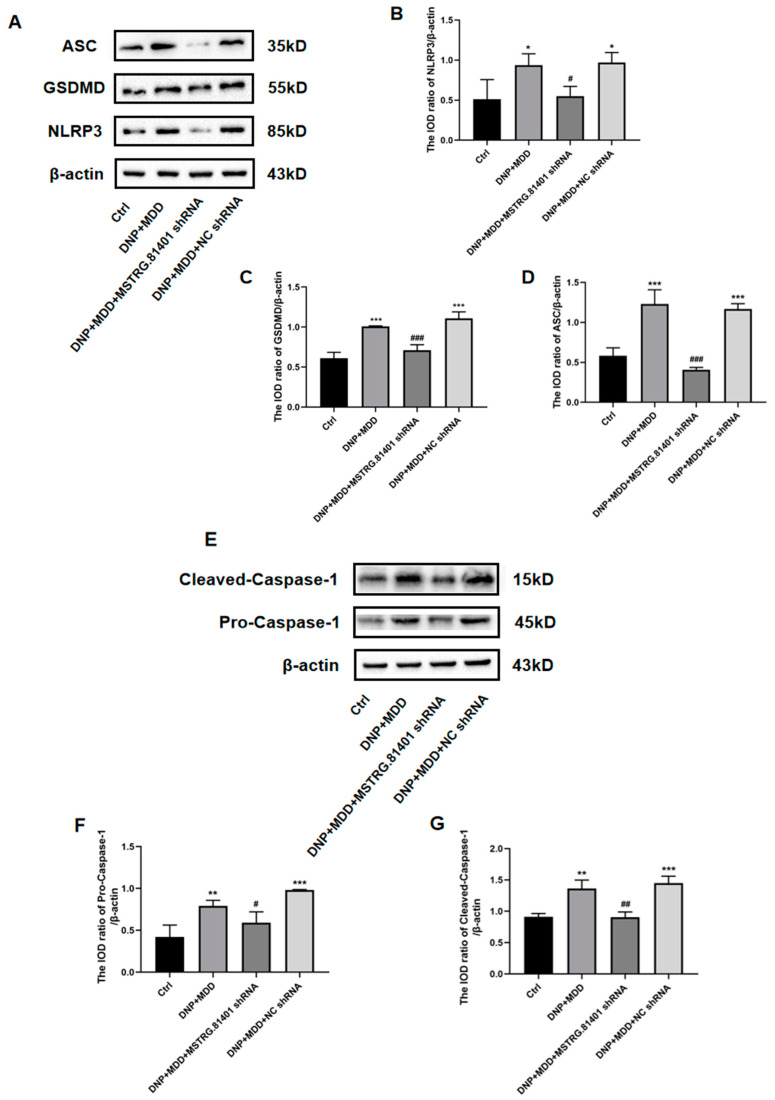
Expression of inflammasome/pyroptosis pathway proteins (**A**–**G**) and inflammatory factors (**H**–**M**) in hippocampus was examined by Western blotting. The bar graphs show the statistical data of relative expression levels of targeted molecules after being normalized by β-actin. NLRP3, GSDMD and ASC (**A**–**D**); procaspase-1 and cleaved-caspase-1 (**E**–**G**); TNF-α (**H**,**I**), IL-1β (**J**,**K**) and IL-18 (**L**,**M**). Values are mean ± SEM, *n* = 6; * *p* < 0.05, ** *p* < 0.01, *** *p* < 0.001 vs. Ctrl group; # *p* < 0.05, ## *p* < 0.01, ### *p* < 0.001 vs. DNP + MDD group.

**Figure 6 ijms-25-01186-f006:**
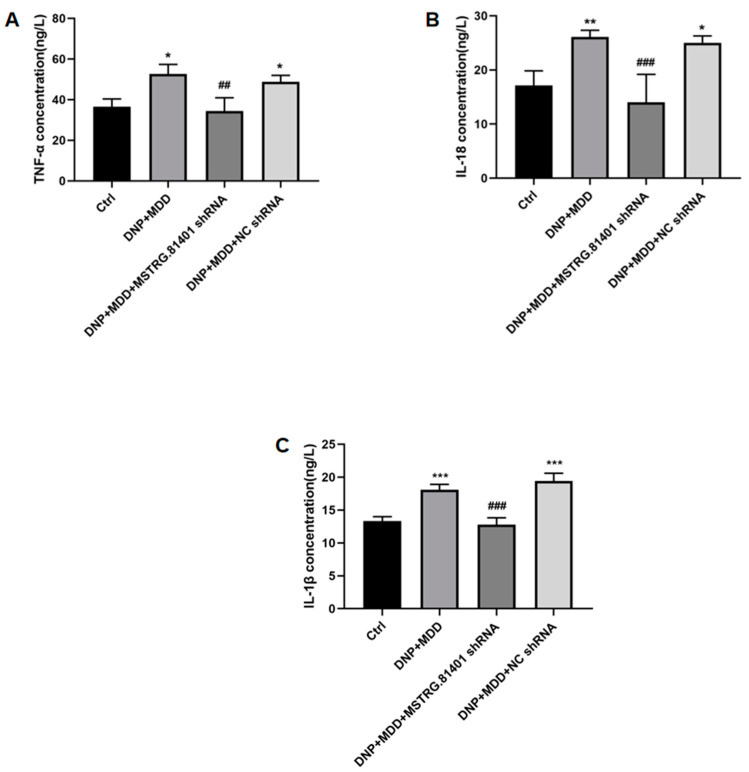
The serum concentrations of TNF-α (**A**), IL-18 (**B**) and IL-1β (**C**) were assessed using ELISA. Values are mean ± SEM, *n* = 6; * *p* < 0.05, ** *p* < 0.01, *** *p* < 0.001 vs. Ctrl group; ## *p* < 0.01, ### *p* < 0.001 vs. DNP + MDD group.

**Figure 7 ijms-25-01186-f007:**
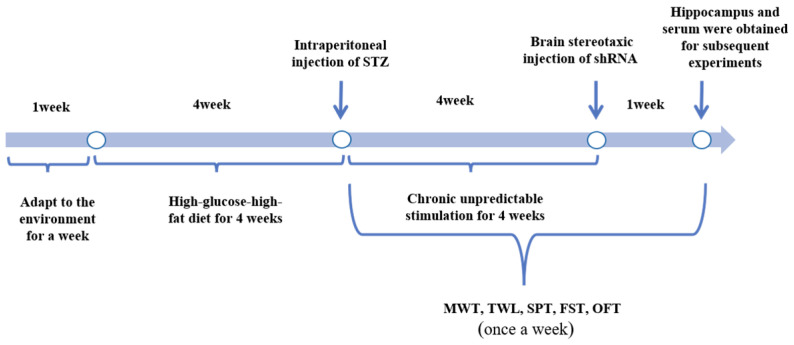
Timeline of the model establishment and sample collection.

## Data Availability

The datasets generated during and/or analyzed during the current study are available from the corresponding author on reasonable request.

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
