# Peer review of "Implication of lncRNA MSTRG.81401 in Hippocampal Pyroptosis Induced by P2X7 Receptor in Type 2 Diabetic Rats with Neuropathic Pain Combined with Depression"

_ijms, 2024, doi:10.3390/ijms25021186_

Round 1
Reviewer 1 Report
Comments and Suggestions for Authors
The manuscript ‘Implication of lncRNA MSTRG.81401 in hippocampal pyroptosis induced by P2X7 receptor in type 2 diabetic rats with neuropathic pain combined with depression‘ analyzes the effect of lncRNA MSTRG.81401 on hippocampal pyroptosis induced by P2X7 receptor in diabetic rats with DNP combined with MDD (DNP+MDD). The study revealed that the expression of lncRNA MSTRG.81401 was significantly elevated in the hippocampus of DNP+MDD rats compared with the control group. The study has shed light on the understanding the role of lncRNA MSTRG.81401 in the regulation of gene expression and cellular processes may provide insights into the molecular mechanisms underlying the observed conditions as well as an insights into these molecular pathways may contribute to the development of targeted treatments for individuals with type 2 diabetes, neuropathic pain, and depression.
After going through the manuscript, I have following comments for the authors.
1. Investigating the interplay between the P2X7 receptor, pyroptosis, and lncRNA MSTRG.81401 in the hippocampus could reveal potential targets for therapeutic interventions. Please discuss this point in the discussion section.
2. Although the rats were randomly divided into four groups, was there any gender bias in the groups?
Comments on the Quality of English LanguageLanguage is fine. Moderate grammatical and syntax errors need to be adjusted
Author Response
Reviewer1
1. Investigating the interplay between the P2X7 receptor, pyroptosis, and lncRNA MSTRG.81401 in the hippocampus could reveal potential targets for therapeutic interventions. Please discuss this point in the discussion section.
Response: We appreciate the valuable suggestion and have elaborated the interplay between P2X7 receptor, pyroptosis, and lncRNA MSTRG.81401 in the revised discussion section. The revised contents are as follows: our study revealed an elevation of lncRNA MSTRG.81401 in the hippocampus of DNP+MDD rats, which subsequently increased the expression and activation of P2X7 receptor. This led to the activation of NLRP3/caspase-1 mediated pyroptosis pathway and subsequent inflammation escalation. Moreover, our experiments demonstrated that the use of shRNA against lncRNA MSTRG.81401 was able to suppress these processes effectively. Therefore, we propose that targeting lncRNA MSTRG.81401 could serve as a promising therapeutic approach for managing diabetic neuropathic pain combined with depression. The modified part is marked in red in revised manuscript (this also applied to the answers to other questions below).
2. Although the rats were randomly divided into four groups, was there any gender bias in the groups?
Response: Considering that the presence of estrogen and the estrus cycle in female mice may influence the outcomes of animal behavior tests, we specifically chose male Sprague-Dawley rats for our study to eliminate any potential effects related to gender difference. We have included a clarification regarding the gender of the rats used in our experiments within the Materials and Methods section. It is worth noting that our previous work has observed varying incidences of diabetic neuropathic pain combined with depression between genders, and we are considering contacting further investigation to explore the underlying mechanisms of this event in the future.
Reviewer 2 Report
Comments and Suggestions for Authors
Please better detail the pyroptosis process for the general medicine reader by an illustrative figure. Please check:Exploring the focal role of pyroptosis in diabetes mellitus. Biointerface Research in Applied Chemistry, 2021, 11(5), 13557-13572. Please detail the potential use of plants for reducing neuropathic pain. please check: Emodin alleviates chronic constriction injury-induced neuropathic pain and inflammation via modulating PPAR-gamma pathway. PLoS One.2023. https://doi.org/10.1371/journal.pone.0287517 and Int. J. Mol. Sci. 2023, 24, 4029. https://doi.org/10.3390/ijms24044029 Please describe the potential use of alpha-lipoic acid for reducing oxidative stress in depression. Please check:Evaluating the efficacy of the treatment with benfotiamine and alpha-lipoic acid in distal symmetric painful diabetic polyneuropathy. Rev. Chim. 2019, 70(9), 3108-14. https://doi.org/10.37358/RC.19.9.7498
Comments on the Quality of English Language
English is ok.
Author Response
We have taken all suggestions by the Reviewer and cited the relevant references in the revised manuscripts, as detailed below.
1.Please better detail the pyroptosis process for the general medicine reader by an illustrative figure. Please check: Exploring the focal role of pyroptosis in diabetes mellitus. Biointerface Research in Applied Chemistry, 2021, 11(5), 13557-13572.
Response: Thanks for the valuable suggestion. We have added a graphical abstract, which can detail the pyroptosis process in the article. In the introduction part, we added that “The caspase-1-dependent formation of plasma membrane pores results in the release of pro-inflammatory cytokines, such as interleukin (IL)-1β and IL-18, leading to cellular inflammation. Emerging evidence has showed pyroptosis plays a vital role in the development of diabetes [17].”
2. Please detail the potential use of plants for reducing neuropathic pain. please check: Emodin alleviates chronic constriction injury-induced neuropathic pain and inflammation via modulating PPAR-gamma pathway. PLoS One.2023. https://doi.org/10.1371/journal.pone.0287517IF: 3.7 Q2 and Int. J. Mol. Sci. 2023, 24, 4029. https://doi.org/10.3390/ijms24044029 IF: 5.6 Q1
Response: Thanks for the valuable suggestion. We have added a paragraph in the introduction section: “Pyroposis can enhance inflammation, thereby facilitating the development of chronic pain [25]. Previous studies conducted in our laboratory as well as others have provided evidence suggesting that certain plant-derived medications, such as emodin, possess the ability to alleviate neuropathic pain through the inhibition of inflammatory responses [26,27].”
3.Please describe the potential use of alpha-lipoic acid for reducing oxidative stress in depression. Please check:Evaluating the efficacy of the treatment with benfotiamine and alpha-lipoic acid in distal symmetric painful diabetic polyneuropathy. Rev. Chim. 2019, 70(9), 3108-14. https://doi.org/10.37358/RC.19.9.7498
Response: We appreciate the valuable suggestion and have added following contents in the introduction section: “Additionally, oxidative stress can induce pyroptosis and promote the development of depression [28]. Oxidative stress and neuroinflammation are closely related to the occurrence of depression, and antioxidants may be an effective drug for treating major depressive disorder (MDD) [29]. For example, alpha-lipoic acid, an antioxidant that can effectively treat distal symmetric painful diabetic neuropathy, may have a potential therapeutic effect on depression [30]. Therefore, direct inhibition of neuroinflammation induced by pyroptosis may be an effective strategy for treating comorbid DNP and MDD. Currently, there is a scarcity of efficacious medications for the treatment of comorbid DNP and MDD. Therefore, it is crucial to explore novel therapeutic agents that have promising inhibitory effects on pyroptosis.”
Reviewer 3 Report
Comments and Suggestions for Authors
A paper on the mechanisms of metabolic dysfunctions + MDD
The design is ok, but before proceeding further, the authors need to insert the ethical number in the methods section.
There is a number at the end, but it is not given by an ethical comitee. It is issued by "Institutional Review Board of Nanchang University, 503 China (NCULAE-20230128089)."
Institutional board can mean anything.
Please explain this first, then we will mouve to science.
Modern science without ethics is nothing!
Author Response
Dear reviewer,
We express our sincere gratitude to the reviewer for pointing out that we were negligent in not including a statement on animal ethical approval in our methodology. We appreciate your reminder, and as per your suggestion, we have now incorporated an Experimental Animal Ethics Statement and provided the Animal Ethics approval number in the methods section of the article. Furthermore, following your valuable input, we have replaced "Institutional Review Board Statement" with Declarations of "Ethics Approval" and "Research Involving Human Participants and/or Animals" at the end of the article. In addition, attached is our stamped ethics statement.
All the changes were marked in red text.
Best Regards,
Yun Gao

Round 2
Reviewer 2 Report
Comments and Suggestions for Authors
The article is much improved. Accept in current form.
Comments on the Quality of English LanguageThe article is much improved. Accept in current form.
Author Response
Dear reviewer,
We sincerely appreciate the reviewer for the critical feedback and insightful comments, which makes our paper more valuable and meaningful.
Best Regards,
Yun Gao
Reviewer 3 Report
Comments and Suggestions for Authors
ok
Comments on the Quality of English Languageok